# Splitting Haploid Chromosomes into Different Nuclei: New Mechanisms of Adaptation in Fungi?

**DOI:** 10.3390/jof11080606

**Published:** 2025-08-21

**Authors:** Lu Liu, James W. Kronstad, Zhongshou Wu

**Affiliations:** 1State Key Laboratory of Rice Biology, Institute of Biotechnology, Zhejiang University, Hangzhou 310058, China; liulu0909@gmail.com; 2Key Laboratory of Molecular Biology of Crop Pathogens and Insects, Zhejiang University, Hangzhou 310058, China; 3The Michael Smith Laboratories, University of British Columbia, Vancouver, BC V6T 1Z4, Canada; kronstad@msl.ubc.ca; 4Department of Microbiology and Immunology, University of British Columbia, Vancouver, BC V6T 1Z4, Canada

**Keywords:** fungal genetics, chromosome segregation, multinucleate cells, evolution and adaptation, pathogenesis, mating, dikaryons

## Abstract

A recent study challenges a fundamental principle of eukaryotic biology that each nucleus houses a complete genome. Two plant pathogenic fungi, *Sclerotinia sclerotiorum* and *Botrytis cinerea*, exhibit a segregated pattern of haploid chromosome distribution across two or more nuclei within each cell. The unequal distribution of the genome between nuclei suggests a coordinated system of internuclear recognition and regulation of cellular functions, a phenomenon previously associated with communication between nuclei of opposite mating type in both ascomycetes and basidiomycetes. Thus, the new study not only shatters expectations about genome biology but also opens new research avenues for understanding fungal adaptation and nuclear behavior.

## 1. Unequal Chromosome Distribution in Nuclei: A Ground-Breaking Discovery in Fungal Biology

Eukaryotic cells are defined by the presence of a membrane-bound nucleus, a feature of cellular organization that enables the compartmentalization of genetic material and the regulation of gene expression. A core principle in cell biology dictates that each nucleus within a eukaryotic cell contains at least one complete set of chromosomes, ensuring genomic integrity during cell division. However, a recent study of two important plant pathogenic fungi, *Sclerotinia sclerotiorum* and *Botrytis cinerea*, has dramatically challenged this dogma [1].

*S. sclerotiorum*, a notorious soilborne pathogen, and *B. cinerea*, a prevalent airborne pathogen, both exhibit multinucleate cells. Previous studies established the haploid chromosome numbers for *S. sclerotiorum* as 16 and *B. cinerea* as 18 core chromosomes plus a variable number of smaller chromosomes in different isolates [2,3,4]. Thus, the expectation was that each nucleus in the binucleate ascospores of *S. sclerotiorum* and the multinucleate conidia of *B. cinerea* would contain a complete haploid genome (a set of chromosomes). Xu et al. [1] provided evidence challenging this expectation by employing a combination of fluorescence microscopy, transmission electron microscopy, fluorescence in situ hybridization (FISH), and single-nucleus PCR to examine the chromosome sets in individual nuclei. They found that each nucleus in *S. sclerotiorum* ascospores contained approximately half the number of chromosomes (approximately 8 of 16) (Figure 1). Furthermore, flow cytometry analysis confirmed that each nucleus contains approximately half the genomic DNA content of a complete haploid genome. Similarly, each nucleus in *B. cinerea* conidia possessed a variable fraction of the 18 core chromosomes [3,4,5,6,7,8] (Figure 1). Crucially, the chromosomal distribution was not uniform; the complement of chromosomes varied between nuclei within the same cell.

To determine the exact composition of chromosomes within individual nuclei, Xu et al. used single-nucleus PCR analysis. They isolated single nuclei, amplified their genomic DNA, and performed PCR with primers specific to each chromosome. The results demonstrated a heterogeneous distribution of chromosomes, reinforcing the idea of an unfixed and variable assortment across nuclei. Importantly, this unfixed pattern (not a consistent “core” arrangement) was observed not only in ascospores and conidia but also in germinating ascospores and older mycelial cells [1]. Overall, these findings raise a number of important questions about mechanisms of chromosome allocation, genome maintenance, and genome reassembly during transitions leading to meiosis and spore formation. Key questions concern the maintenance of genome integrity and fidelity, and Xu et al. propose that nuclei somehow remain paired, perhaps by connecting filaments or membranes (Figure 1A), to allow nuclear recognition and coordination.

## 2. Connections with Fungal Adaptation, Mating Type, and Pathogenesis

The findings of Xu et al. have a number of implications for fungal biology. First, “genome integrity” in eukaryotic cells has traditionally been viewed as ensuring accurate chromosome segregation to maintain genetic fidelity. However, fungi such as *S. sclerotiorum* may tolerate certain levels of genomic variability or controlled mis-segregation. Such flexibility could be an adaptive mechanism to promote genetic diversity rather than preserving uniformity. They might be geared toward guaranteeing that each nucleus retains a core set of essential genes, while allowing for variation in other chromosomal contents (Figure 1A). This arrangement could facilitate a balance where vital genetic information is conserved, yet some degree of heterogeneity is tolerated and even advantageous. Such genome plasticity aligns with the broader concept of “evolvability,” where organisms leverage controlled genome instability to generate heritable variation that enhances survival and evolutionary potential, as exemplified in other systems where non-canonical segregation or instability proves beneficial [5,6]. Experimental approaches such as single-nucleus RNA-seq could test this idea by comparing transcriptional profiles of nuclei with different chromosome complements, while competition assays under stress conditions could determine whether chromosomal variability provides a fitness advantage.

On the other hand, this new chromosome assortment system generates a dynamic pool of genetic variability within single cells. By partitioning a limited and variable subset of chromosomes into individual nuclei, these fungi might reduce the genetic load associated with deleterious mutations, thus lowering the fitness costs typically linked to genome instability. Additionally, this reduction may allow for a broader exploration of the genetic landscape, facilitating the emergence of advantageous gene combinations. This mechanism can potentially accelerate the accumulation and dissemination of virulence factors, antibiotic resistance genes, or stress-response elements, culminating in rapid phenotypic shifts that enhance pathogenicity and survivability (Figure 1B). To investigate this possibility, virulence assays comparing wild-type strains to artificially stabilized variants could reveal whether chromosomal variability enhances pathogenicity, while comparative genomics of field isolates might correlate specific chromosome distributions with host adaptation. Parallel systems in other organisms, such as cancer cells exhibiting chromosomal instability [7] or bacteria with dynamic plasmid exchange, further support the idea that genome flexibility enhances adaptive capacity [8,9]. In fungi, the ability to generate diverse genotypic combinations in situ, even alongside periodic sexual reproduction, presents a potent strategy for rapid evolution [10]. *S. sclerotiorum* relies on annual sexual cycles for propagation, and *B. cinerea* frequently undergoes sexual reproduction in nature; vegetative genotypic variability may contribute to short-term adaptation and population diversity. This mechanism could operate in parallel with sexual cycles, particularly in environments where rapid responses to selective pressures are advantageous. Therefore, the chromosomal segregation system is not merely a cellular curiosity but a fundamental evolutionary tool that fuels the continuous diversification and adaptability of these pathogens, enabling them to thrive amid the shifting challenges of their environments.

The mechanism underlying nuclear recognition between nuclei containing irregular chromosome complements remains unknown, though insights may be drawn from well-characterized mating-type systems in other fungal species. In many ascomycetes and basidiomycetes, nuclear pairing during karyogamy and meiosis is mediated by mating-type loci through sophisticated recognition mechanisms [11,12,13,14]. In heterothallic species like *Schizophyllum commune*, pheromone/receptor systems establish nuclear recognition zones through MATB loci [15,16], while in *Podospora anserina*, mating-type transcription factors (FPR1/FMR1) regulate similar pathways [11,17,18]. However, *S. sclerotiorum*’s homothallism and *B. cinerea*’s heterothallic MAT1 locus structure suggest these canonical mechanisms may not apply [3]. Instead, the filaments or membrane structures hypothesized by Xu et al. to link nuclei may function for internuclear recognition (Figure 1A and Figure 2B). In principle, such inter-nuclear connections could involve extensions or modifications of the nuclear envelope, mediated by proteins analogous to those found in the linker of nucleoskeleton and cytoskeleton (LINC) complex, which bridges the nuclear interior and cytoplasmic cytoskeleton in many eukaryotes [19]. In fungi, SUN-domain proteins (e.g., Sad1 in *Schizosaccharomyces pombe*) and KASH-domain proteins (e.g., Kms1, Kms2) are known to anchor nuclei to cytoskeletal elements and to each other during meiosis and nuclear migration [20]. Other nuclear envelope-associated proteins, such as components of the spindle pole body (e.g., Mps3 in *Saccharomyces cerevisiae*) or nuclear pore complex proteins (nucleoporins), could also contribute to stable nuclear positioning and recognition [20]. Actin- or microtubule-based tethers, possibly crosslinked by motor proteins (dynein, kinesin), represent another plausible mechanism [21]. Identifying fungal homologs of these protein families in *S. sclerotiorum* and *B. cinerea*, followed by localization studies using fluorescent tagging, could help determine whether similar molecular machinery underlies the hypothesized nuclear pairing.

The coexistence of nuclei with incomplete but complementary chromosome sets raises the question of how overall cellular function is coordinated. One possibility is that transcripts and proteins produced by different nuclei are freely exchanged through the shared cytoplasm, allowing the pooled gene products to meet the needs of the whole cell. In filamentous fungi, mRNAs and even ribonucleoprotein complexes are known to traffic between hyphal compartments via cytoplasmic streaming and septal pores, and inter-nuclear complementation of gene products has been observed in heterokaryons [22]. Another possibility is active internuclear signaling, where paired nuclei coordinate gene expression through shared regulatory factors, potentially mediated by nuclear pore complex-associated transport or signaling cascades linked to cytoskeletal connections [22]. Such mechanisms could compensate for incomplete genomic content in individual nuclei, ensuring that essential cellular processes proceed normally despite chromosome-level variability. Live-cell imaging of fluorescently tagged transcripts or proteins, coupled with disruption of cytoplasmic continuity, could test the degree to which molecular exchange underpins functional coordination.

It is important to note that internuclear recognition and aspects of coordinated nuclear partitioning are relevant for the ability of some phytopathogenic fungi to cause disease. A fascinating example comes from nuclear coordination in *Magnaporthe oryzae* where one of the three nuclei in the germinating conidium on a leaf surface moves into the germ tube and undergoes mitosis with subsequent movement of one nucleus back to the conidium for eventual autophagic destruction [23,24]. The other mitotic nucleus migrates into the appressorium for eventual entry into the penetration hypha [23,24]. For basidiomycete pathogens, observations on the virulence of different cell types of *Ustilago maydis* support the importance of nuclear coordination and recognition of the nuclear state. In this pathogen, mating between haploid cells results in a filamentous, dikaryotic cell type that proliferates in maize tissue to cause tumors [25]. There is a long-standing observation that inoculating plants with diploid strains results in less severe disease compared with dikaryons established by mating [25,26,27]. Furthermore, normally non-pathogenic haploid strains can be engineered to cause disease by introduction of mating-type genes of opposite specificity, and these strains are also less virulent than dikaryons [26,27]. The underlying mechanism(s) for the benefits of being dikaryotic are unknown, but these observations support the idea that internuclear recognition can contribute to fungal pathogenesis. For *S. sclerotiorum* and *B. cinerea*, live imaging of GFP-tagged nuclei during host infection could reveal whether particular nuclear states dominate during different infection stages, while targeted disruption of putative nuclear pairing mechanisms might test their importance for disease development. These approaches would help determine whether the newly discovered chromosome distribution system represents another example of nuclear behavior contributing to fungal pathogenicity.

## 3. Concluding Remarks and Future Directions

Overall, the findings of Xu et al. challenge the conventional understanding of eukaryotic nuclear structure and function, highlighting a previously unappreciated flexibility in chromosome partitioning. Notably, *S. sclerotiorum* and *B. cinerea* are closely related within the same taxonomic family, the Sclerotiniaceae [3], but the phenomenon they exhibit may not be unique to this lineage. Many other filamentous ascomycetes (e.g., *Fusarium*, *Neurospora*, *Aspergillus*) and some basidiomycetes (e.g., *Ustilago*, *Coprinopsis*) also possess multinucleate hyphae or spores, yet single-nucleus chromosome complements in these organisms have not been systematically examined. Given that most cytological studies assume complete haploid or diploid sets in each nucleus, partial chromosome complements could have been overlooked due to limitations in resolution or the absence of targeted single-nucleus analyses. Moreover, certain multinucleate animal and plant cells—such as syncytial endosperms or skeletal muscle fibers—might, in principle, display analogous partitioning, although this remains to be tested. A systematic survey across diverse multinucleate organisms using high-resolution imaging and single-nucleus genomics could reveal whether this unusual chromosome segregation strategy is more widespread than currently appreciated.

The findings for *S. sclerotiorum* and *B. cinerea* pave the way for various new research directions. For example, the unfixed arrangement of chromosomes presents exciting possibilities for genetic engineering of these organisms. The simpler genome structure, effectively halved in each nucleus, could facilitate the construction of targeted gene deletions and other genetic manipulations. Notably, Xu et al. observed unexpected mutant behaviors from their initial forward genetic screen that led to the discovery of the novel pattern of chromosome distribution [1]. Therefore, this research not only confronts existing paradigms but also opens the door to more facile genetic analysis and a reimagining of how genetic variability might fuel environmental adaptability in pathogenic fungi and possibly other eukaryotes.

Further investigation is needed to uncover the mechanisms regulating the unique chromosome segregation in *S. sclerotiorum* and *B. cinerea*. A comprehensive study of the processes involved in mitotic and meiotic divisions is necessary to understand how they ensure genetic integrity in multinucleate cells, as well as the identification of specific genes influencing these mechanisms. A dedicated investigation is needed to search for the filaments or membrane structures proposed by Xu et al. to connect paired nuclei and maintain genome fidelity during mitotic divisions [1] (Figure 1 and Figure 2). The interconnectedness of this hypothesis for pairing with the biological significance of segregated chromosomes is noteworthy. Understanding how these proposed structures function and how these bi-nuclei/multi-nuclei communicate could reveal a novel mechanism of nuclear recognition and help establish why these fungi developed such a mechanism despite possessing membrane-bound nuclei. In addition, it would be particularly worthwhile to explore whether there is a preferential pattern retained across nuclei, forming a conserved “core genome” that ensures essential cellular functions. Such a core could include chromosomes carrying housekeeping genes, primary metabolic pathways, or key virulence factors, while other chromosomes may vary more readily to generate diversity.

## Figures and Tables

**Figure 1 jof-11-00606-f001:**
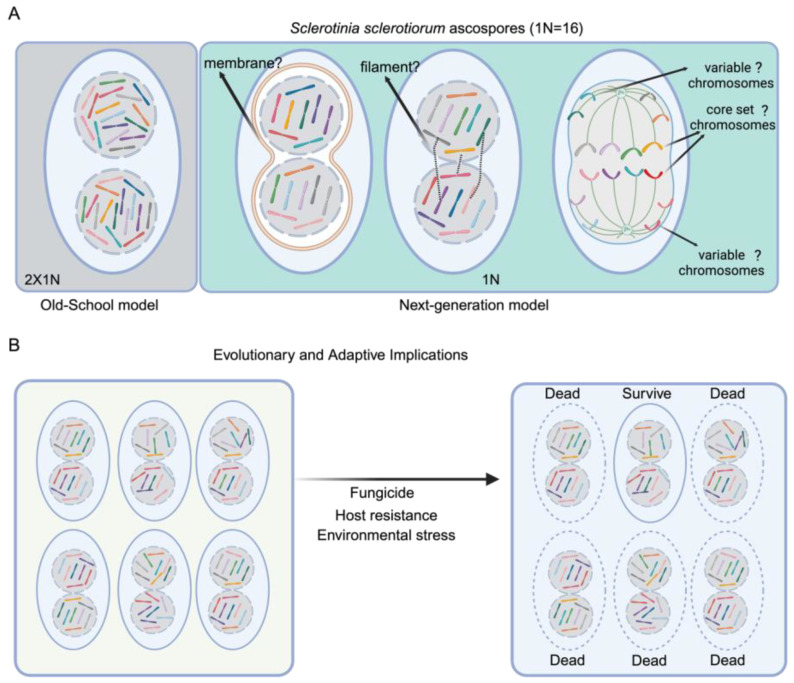
Schematic representation of chromosome distribution in *S. sclerotiorum* ascospores and evolutionary and adaptive implications. (**A**) The Old-School model shows that each nucleus contains a full set of 1N chromosomes in multinucleate fungal cells. However, the current study finds that chromosomes are segregated in multi-nuclei in an unfixed manner. The mechanisms ensuring genome fidelity during mitosis-such as dynamic filaments or membrane structures, remain hypothetical and untested. Another possibility is that a core set of chromosomes is retained in each nucleus, while others are variably distributed. (**B**) Diverse genotypic profiles through unfixed chromosome sets lead to rapid adaptation to environmental stress, fungicide, and host resistance.

**Figure 2 jof-11-00606-f002:**
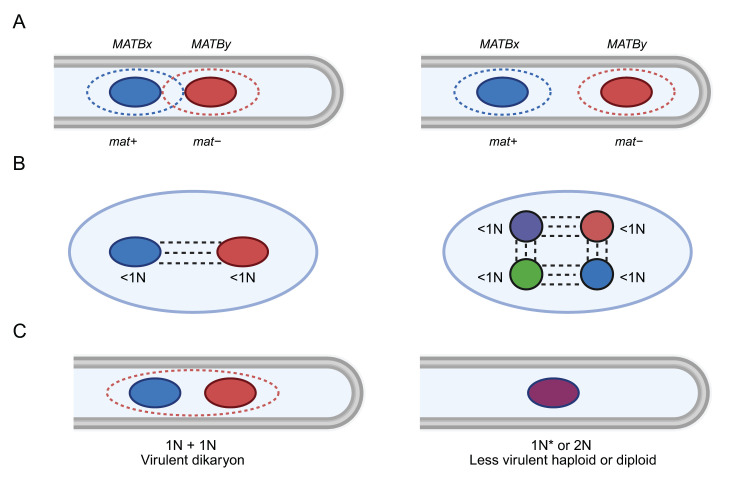
Models for internuclear recognition and potential contributions of mating type. (**A**) A depiction of the model involving pheromone and pheromone receptors establishing zones of influence/identity around nuclei, as proposed by [11,16]. The filamentous cell on the left depicts the situation where mating type loci (e.g., *mat*+ and *mat*− in *P. anserina* or *MATB* alleles of different specificity x and y in *S. commune*) create zones that facilitate recognition when nuclei are in close proximity. The cell on right depicts the lack of recognition when nuclei are far apart. (**B**) A model in which connections (e.g., membranes, proteins, cytoskeleton) enable recognition among nuclei for ascospores of *S. sclerotiorum* on the left and conidia of *B. cinerea* on the right. (**C**) An illustration of the difference in virulence for the dikaryotic cell type relative to the diploid cell type or a haploid strain carrying both specificities of the mating type loci (1N*) of *U. maydis*.

## Data Availability

Not applicable.

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
