# Peer review of "Splitting Haploid Chromosomes into Different Nuclei: New Mechanisms of Adaptation in Fungi?"

_jof, 2025, doi:10.3390/jof11080606_

Round 1

Reviewer 1 Report

Comments and Suggestions for Authors

This is a commentary on the recently published Science article by Xu et al (2025. Science. 388 (6748):784-8). Unfortunately, it reads more as an extension of the original paper’s discussion rather than a scientific evaluation of the data and implications of the data on the biology of these fungi. 

L92-96. This discussion of the role that dynamic vegetative genotypic variability may play in evolution "without the need for long-term sexual cycles", ignores the fact that S. sclerotiorum frequently (annually) completes a sexual cycle and is dependent on the sexual cycle to propagate the organism as it does not produce asexual spores. Additionally, B. cinerea frequently completes its sexual cycle in the environment and is routinely produced under laboratory conditions with mating type being the only observed barrier to sexual compatibility. As such, how vegetative genotypic variation relates specifically to the fungi under study here is not obvious.

L100-129. This paragraph begins by hypothesizing that mating type loci may be involved in internuclear recognition, it continues by providing some examples of gene expression influenced by nuclear positioning and concludes with the acknowledgement that mating type is unlikely to be involved in internuclear communication in the context of non-haploid nuclear recognition in B. cinerea and S. sclerotiorum. This paragraph does not seem to add value or new insights toward explaining how nuclei of complementing chromosomal makeup may recognize each other and stay paired, other than to say, it is probably not through mating-type specific gene expression. This paragraph could be summarized in one sentence.

L38. Instead of "…Xu et al. [1] demonstrated that this expectation is incorrect" the authors should state "... Xu et al. [1] provide evidence challenging this expectation..."

Reviewer 2 Report

Comments and Suggestions for Authors

This is a well-structured and intellectually stimulating commentary that explores a paradigm-shifting discovery in fungal biology, the unequal distribution of haploid chromosomes across multiple nuclei within a single cell. The manuscript is scientifically compelling, well-referenced, and written in a manner that integrates cutting-edge research with broader implications for genome integrity, fungal evolution, and cellular behavior. It is particularly commendable in highlighting the relevance of internuclear coordination and pathogenesis and in identifying the broader significance for nuclear biology.

Minor Revisions and Suggestions:

  1. At times, the terms “haploid genome,” “partial genome,” and “unfixed chromosomes” are used interchangeably. It may help the reader if the authors briefly define these terms upfront or ensure consistent usage throughout.
  2. The shift from genome segregation discussion to mating-type influence could benefit from a clearer transitional sentence explaining why mating-type mechanisms are being introduced in this context.
  3. In several places, Xu et al. [1] is referenced multiple times within the same paragraph. Consider streamlining to avoid redundancy while ensuring that attribution is clear.

Questions for Authors

  1. Do the authors believe that certain essential chromosomes are preferentially retained across nuclei (i.e., a conserved "core genome")? Could this be a subject of future empirical study?
  2. Could the authors elaborate more specifically on what kind of membrane structures or proteins might be responsible for the hypothesized nuclear pairing? Are there any candidates from existing nuclear envelope studies in fungi?
  3. While the focus is on S. sclerotiorum and B. cinerea, could the authors speculate on whether this mechanism might be overlooked in other multinucleate fungi or even outside Sclerotiniaceae?
  4. The potential for easier genetic manipulation is briefly noted. Could the authors expand slightly on how this might be practically implemented in a research or biotechnological setting?
  5. If nuclei within the same cell harbor different chromosomes, how is cellular function coordinated? Are there known mechanisms (e.g., shared cytoplasmic factors or inter-nuclear signaling) that could compensate for the incomplete genomes?

Reviewer 3 Report

Comments and Suggestions for Authors

This review discusses the spectacular discovery by the research group of Prof. Xin Li, that Ascospores of Sclerotinia sclerotiorum and conidia of the closely related Botrytis cinerea contain nuclei with fragmented chromosome content, the whole haploid genome being divided between two (Ss ) or more (Bc) haploid nuclei. The review is very nicely and clearly written, it describes the consequences of this newly discovered segregation of nuclear genetic information and outlines possible evolutionary consequences. The review contains a nice illustration which helps with understanding the phenomenon of this new type of segregation and some of its evolutionary implications.

I have only a few comments and suggestions:

L35. The authors write 'Previous studies established the haploid chromosome numbers for these fungi as 16 and 18, respectively [2-4].' This is correct for S. sclerotinia (16) and for B. cinerea strain B05.10 (18 chromosomes). What souldbe mentioned here is that in all known Botrytis and Sclerotinia spp., the number of core chromosomes, namely 16, is highly conserved. In addition, B. cinerea isolates contain a variable number of small chromosomes which contain relatively few non-essential genes of unknown function (Valero-Jimenez et al., 2020, Genome Biol Evol 12(12):2491-2507). For comparative analysis of chromosome segregation in Sclerotinia and Botrytis, one would focus mostly on these 16 core chromosomes.

2. The Chapter 'Connections with fungal adaptation, mating type and pathogenesis' is well written, however, it would profit if the authors might propose or outline experimental strategies which could test some of the hypotheses that have been raised in the original paper by Xu et al. and in this review.

3. In Fig. 1, the two figure plates have been labeled with A and B, whereas in the legend, the are refered to as 'Top panel' and 'Lower panel', which should be substituted by 'A' and 'B'.

Round 2

Reviewer 1 Report

Comments and Suggestions for Authors

I previously reviewed a version of this manuscript and feel that my major concerns have been addressed.

In the current version of this manuscript I am confused by the statements made in L335-343: "In practical terms, the reduced chromosome comple- ment in each nucleus could simplify genetic manipulation by decreasing the number of alleles that need to be simultaneously targeted, thereby improving the efficiency of ho- mologous recombination or CRISPR/Cas-mediated editing. For example, introducing mu- tations in essential genes could be achieved in nuclei lacking redundant alleles, reducing the likelihood of functional compensation and allowing phenotypes to manifest more readily. In biotechnology, this system could be exploited to engineer industrial fungal strains with modular chromosomal toolkits, where specialized nuclei within heterokary- ons perform distinct metabolic functions." The authors use the term "alleles" but in the context of haploid fungi derived from a single ascospore, alleles do not exist. Further, how the efficiency of homologous recombination or CRISPR/Cas mediated genomic engineering might be improved in strains harboring nuclei with partial genomes is not clear. In addition the meaning of the deletion of essential genes "in nuclei lacking redundant alleles" is not understood by this reviewer. This section of the manuscript could be easily deleted without changing the main points made by the authors. Otherwise this section needs be rewritten to be better understood by readers.

Author Response

We thank the reviewer for this helpful feedback and for pointing out the potential confusion regarding the use of the term “alleles” and the interpretation of genome editing efficiency in this context. To avoid ambiguity and improve clarity, we have removed this section from the revised manuscript.